# Quality over Quantity: The Association Between Daily Social Interactions and Loneliness

**DOI:** 10.3390/ijerph22091411

**Published:** 2025-09-10

**Authors:** Siyun Peng, Adam R. Roth

**Affiliations:** 1School of Aging Studies, University of South Florida, Tampa, FL 33612, USA; 2Department of Sociology, Indiana University, Bloomington, IN 47405, USA; 3Irsay Institute, Indiana University, Bloomington, IN 47405, USA; 4Department of Sociology, Oklahoma State University, Stillwater, OK 74078, USA; adam.roth@okstate.edu

**Keywords:** loneliness, social interactions, social isolation, ecological momentary assessment, social strain

## Abstract

Understanding what aspects of daily life protect against loneliness is increasingly important for promoting well-being among older adults. This study investigates how different types of everyday social interactions are associated with chronic loneliness. We analyze data that were collected via in-home surveys and an ecological momentary assessment module from a probability-based sample of 272 adults aged 55 and older residing in Indiana. Participants were prompted four times per day over the course of seven days to report on their current activities and social experiences. Contrary to common assumptions, the quantitative measures of daily social interactions, such as the proportion of moments spent alone, the proportion of moments spent socializing, and the mean number of interaction partners during the study period were not significantly related to loneliness. However, the qualitative measures of social interactions—specifically, the presence of bonding social capital (e.g., emotional closeness) and the absence of stressful interactions—were associated with lower levels of loneliness. These findings underscore the importance of emotionally meaningful engagement and social strain over the sheer frequency or quantity of interactions, suggesting that the quality of daily social experiences is a key factor in supporting mental well-being in later life.

## 1. Introduction

Loneliness is a distressing emotional state that arises when individuals perceive a discrepancy between the social relationships they desire and those they actually have [1]. While it is often associated with social isolation—the objective lack of social contact—the two are conceptually and empirically distinct [2]. Some individuals may feel lonely even if they are routinely surrounded by others, while others with limited contact may not feel lonely at all. Understanding how social contact relates to loneliness has been a central concern in public health research, particularly because loneliness in later life has been linked to serious health risks, including cardiovascular disease, depression, cognitive decline, and increased mortality [3,4,5,6,7].

Prior studies have shown that both the quantity of social interactions (e.g., frequency of contact or social isolation) and the quality of those interactions (e.g., perceived positivity or closeness) can shape the experience of loneliness [8,9,10,11,12]. This association has been documented across nations and cultures [13,14,15]. In the present study, we focus on investigating how aggregate measures of daily social contact are associated with chronic or trait loneliness, as measured by the UCLA Loneliness Scale during an in-person survey [16], rather than with momentary loneliness assessed through ecological momentary assessment (EMA). We achieve this by combining data that were collected via an in-home survey and an EMA module to examine and compare how a broad range of moment-to-moment social experiences relate to perceptions of chronic loneliness in later life.

### 1.1. Dimensions of Social Interaction

Social interactions—which refer to the momentary exchanges between two or more people—have two broad dimensions. The first dimension examined in this study is the quantitative dimension, which includes frequency of contact and social isolation. These objective indicators have been consistently linked to loneliness [5,17,18]. Individuals with limited social contact or who experience social isolation often lack opportunities for meaningful engagement, increasing the risk of emotional disconnection and loneliness.

One critique of quantitative measures is that not all social interactions are equally effective in alleviating loneliness [8,9,10]. Therefore, it is essential to also consider the second, qualitative dimension. Social interactions offer the chance for people to build social capital—a term that refers broadly to the beneficial resources exchanged between a group of people [19,20]. We focus on two forms of social capital—bonding and bridging. Bonding social capital refers to the emotional and relational resources—such as social support, a sense of belonging, and mutual trust—that are typically embedded in strong ties with similar others. These elements are widely recognized as central protective factors against loneliness [12,21,22]. Bridging social capital, by contrast, refers to access to novel information, resources, and environment that typically derive from weak ties to dissimilar others [23]. While bridging social capital is often linked to outcomes such as job opportunities [24], cognitive health [25,26], and general health [27], it may also buffer loneliness by expanding one’s social world and offering alternative pathways to a sense of social connection.

However, it is important to recognize that not all social interactions are supportive or beneficial. Frequent contact can increase the risk of conflict, tension, or ambivalence—forms of social strain that can exacerbate loneliness. Conflict and strain have been linked to greater loneliness across various contexts [28,29,30]. Accordingly, this study treats social strain as a key qualitative dimension and examines how older adults’ perceptions of negative social interactions contribute to their overall experience of loneliness.

Thus, we hypothesize:

**H1:** *Higher frequency of contact is associated with lower chronic loneliness*.

**H2:** *Higher levels of bonding and bridging social capital are associated with lower chronic loneliness*.

**H3:** *Higher levels of stressful interactions are associated with higher chronic loneliness*.

### 1.2. The Current Study

Leveraging EMA data to measure different dimensions of social interaction offers a more nuanced and ecologically valid approach to understanding chronic loneliness. Traditional retrospective surveys often fail to capture the variability and context of daily social experiences whereas EMA allows for real-time reporting of both the quantity and quality of interactions as they naturally occur [31]. This method enables researchers to distinguish between superficial contact and emotionally meaningful exchanges, account for both supportive and strained interactions, and detect subtle patterns that may be missed in global assessments. By capturing moment-to-moment fluctuations in social experiences, EMAs provide critical insight into how different types of interactions—social isolation, bonding, bridging, and negative encounters—contribute to the variation in chronic loneliness in later life.

## 2. Methods

### 2.1. Sample

Data are from the Social Environment and Cognitive Health in Urban and Rural Areas (SECHURA) study, a state-representative sample of Indiana residents aged 55 or older. During November 2023 and March 2024, the SECHURA study conducted a computer assisted in-person interview (CAPI) survey with 509 participants (61% response rate). Each CAPI survey was followed by one week of EMA collection via smartphones. 272 of the 509 SECHURA respondents (53.4%) participated in the EMA module. For the following seven days, EMAs were administered four random times per day between 8:00 and 10:00 am and noon to 8:00 pm by “pinging” participants through smartphone notifications. Each notification was spaced at least two hours apart. These four different windows aimed to capture a representative sample of the private and public spheres where participants spend their time. The EMA questions asked participants to provide real-time reports about where they were (setting), what they were doing (activity), and whom they were with (social tie) and to assess their cognitive and emotional well-being. Participants had 20 min to respond to each notification with three additional reminder notifications being sent in five-minute intervals. The vast majority of participants (84%) completed their EMAs in less than two minutes with an average time of one minute and 18 s. The 272 (53.4%) participants who participated in the EMA module produced a total of 5766 EMAs, which yield a compliance rate of 76%. Sampling weights were provided by the SECHURA to address sampling and response rates. Participants who did not own a smartphone (*n* = 9) were offered a loaner phone to reduce selection bias. Internet access was only required during the installation phase of the smartphone app. Further details on the SECHURA research design are documented elsewhere [32]. The SECHURA study is conducted in accordance with the Declaration of Helsinki and approved by the IRB of Indiana University (#2008142667). It is not preregistered. Further details on the SECHURA research design can be found elsewhere [32].

### 2.2. Measures

*Loneliness* was measured using the average score of the 3-item UCLA loneliness scale (Cronbach’s α = 0.82) [16]: “How often do you feel that you lack companionship?”, “How often do you feel left out?”, and “How often do you feel isolated from others?” Response options were: 1 = Hardly ever (or never), 2 = Some of the time, and 3 = Often. Higher scores indicate greater levels of chronic loneliness.

### 2.3. Quantitative Measures of Social Interactions

*Social isolation* was assessed using *the proportion of EMAs spent alone*. During each EMA, participants were asked: “When you heard the notification, who were you with?” Responses were coded as “alone” if the participant selected “nobody.” Each participant’s proportion of EMAs spent alone was calculated by dividing the number of EMAs where they reported being alone by their total number of completed EMA responses.

Another quantitative indicator of social interaction was *the proportion of EMAs spent socializing*. During each EMA prompt, participants were asked: “What were you doing when you heard the notification?” Response options included: “eating/drinking,” “socializing,” “relaxing,” “working,” “shopping,” “household chores,” “volunteering,” “transporting (e.g., car, bus, bike),” “medical care,” and “other.” Responses coded as “socializing” were used to calculate the proportion of EMAs spent socializing. This proportion was calculated by dividing the number of EMA where participants were socializing by their total number of completed EMA responses.

The final quantitative measure of social interaction was the average number of interaction partners. In each EMA prompt, participants were asked: “Approximately how many people were you interacting with when you heard the notification?” To reduce the influence of outliers, responses indicating more than 10 people (1.4% of all EMA responses) were collapsed as 10. The mean number of interaction partners was calculated for each participant by averaging their responses across all completed EMA entries.

In addition to EMA-based measures of social interaction quantity, we included personal network size—a commonly used survey-based indicator of social interaction quantity [5]. This measure serves as a sensitivity check to replicate the associations between survey-level quantitative social interaction indicators and loneliness frequently reported in prior research [5,17,33]. We measured personal networks by using *name generators* to elicit names of individuals who were activated in the past 6 months without capping the total number of names that could be listed. Four name generators elicited discussants about important matters (i.e., with whom you discussed personal problems), health matters (i.e., with whom you discuss your health, or you can really count on for help when you have physical or emotional problems), health regulators (i.e., who are always talking about your mental and physical health and trying to get you to do things about them?), and regular contacts (i.e., with whom you regularly spend your free or leisure time) based on the PhenX Social Network Battery [34]. We calculated the *network size* based on total names listed in those four name generators.

### 2.4. Qualitative Measures of Social Interactions

*Bonding social capital* was measured using two items. For every EMA during which participants reported being with someone, they were asked to rate the following on a 1–10 scale: (1) “I felt a close, personal connection during this interaction” and (2) “My mood was good because of this interaction”. The scale demonstrated acceptable consistency (Cronbach’s α = 0.71), supporting its reliability as a unified measure. Bonding scores were calculated for each participant by averaging their responses across all completed EMA entries.

*Bridging social capital* was measured using three items. At each EMA prompt, participants rated the following on a 1–10 scale: (1) “I was being exposed to new information or recommendations,” (2) “I was being challenged to consider a new perspective,” and (3) “I was using a lot of cognitive effort”. From a social network perspective, it is the resources and access to novel, non-redundant information and experiences that individuals gain through these weak ties—not the ties themselves—that constitute bridging social capital [24,35]. Our measurement, which captures exposure to new information, reflects this theoretical emphasis on the resources and opportunities provided by bridging connections, consistent with prior conceptualizations in the literature. Because this study focused specifically on social interactions, ratings were only included when participants reported interacting with other people. For example, while individuals may encounter new information while watching videos, such instances were excluded, as they do not involve interpersonal exchange and thus do not reflect bridging social capital. The scale demonstrated high internal consistency (Cronbach’s α = 0.88), supporting its reliability as a unified measure. Bridging scores were calculated for each participant by averaging their responses across all completed EMA entries.

In addition to assessing positive social interactions, *stressful interactions* were measured using the item: “I felt stressed because of this interaction.” Responses were rated on a scale from 1 (No stress at all) to 10 (Very stressed). Stressful interactions scores were calculated for each participant by averaging their responses across all completed EMA entries.

There were 11 participants who never reported being with someone during the observation period; therefore, they were missing on bonding social capital, bridging social capital, and stressful interactions.

### 2.5. Covariates

The following covariates were included in the final models. *Age* was measured in years. *Gender* was measured as 0 = men and 1 = women. *Race* was measured as 0 = Non-White or 1 = White due to majority of participants were White in Indiana. *Education* was measured as 1 = less than high school, 2 = high school or GED, 3 = some college/technical school, and 4 = college and above. *Employment status* was measured as 0 = not working and 1 = currently working. *Functional activity limitation* was measured using a 10-item scale (e.g., assembling tax records, buying groceries, heating water, preparing a meal, traveling out, etc.) [36]. The responses were recoded as 0 for no limitations across all 10 items and 1 for any limitation in any of the 10 items. To accurately assess the relationship between the EMA measures of social interactions and loneliness, it is important to adjust for the overall proportion of completed EMA surveys, as differences in compliance may bias estimates of social interaction and its association with loneliness.

### 2.6. Analytic Strategy

Because loneliness was measured at the participant level, all EMA-based social interaction measures were aggregated to the participant level to align with the level of analysis. Linear regression models were used to examine predictors of loneliness, adjusting for age, gender, race, education, employment status, and functional activity limitations. Proportion of completed EMA was included in all models that have EMA measures of social interactions. Each social interaction measure—proportion of time spent alone, proportion of time spent socializing, mean number of interaction partners, bridging social capital, bonding social capital, stressful interactions, and network size—was entered into separate models as an individual predictor of loneliness.

All social interaction measures and loneliness were z-standardized prior to inclusion in the models to facilitate comparison of effect sizes. Survey weights were applied to account for sampling design and differential response rates. No missing data were observed for covariates. Statistical significance was set at *p* < 0.05. We examined the residuals from survey-weighted regression models and found them to be approximately normally distributed. Given the use of robust variance estimation, the analyses are robust. All analyses were conducted using Stata 19. Replication code is available online at https://github.com/PengSiyun/EMA-and-loneliness (accessed on 1 September 2025).

## 3. Results

Table 1 presents descriptive statistics for our sample. The average age was 66.10 years (SD = 7.70), with participants ranging from 55 to 88 years old. Approximately half of the sample were women (51%) and the majority identified as White (87%). Regarding educational attainment, 43% held a college degree, while smaller proportions had completed some college (24%), a high school diploma or GED (27%), or less than high school (5%). Forty percent of participants were employed.

### 3.1. Dimensions of Social Interaction Linked to Loneliness

Figure 1 shows that EMA-based quantitative aspects of social interaction—proportion of EMAs spent alone (β = 0.09, *p* > 0.05), the proportion of EMAs spent socializing (β = 0.07, *p* > 0.05), and the mean number of interaction partners across EMAs (β = 0.02, *p* > 0.05)—were not significantly associated with loneliness. However, network size, a survey-based quantitative measure of social interaction, was related to loneliness (β = −0.13, *p* < 0.05).

In contrast, qualitative dimensions of social interaction showed significant associations. Lower levels of bonding social capital (β = −0.24, *p* < 0.001) and higher levels of stressful interactions (β = 0.26, *p* < 0.001) were both linked to greater loneliness. However, no significant association was observed for bridging social capital (β = 0.15, *p* > 0.05). Appendix A provides the full details of the models depicted in Figure 1.

### 3.2. Sensitivity Analysis

Eleven participants reported no social interactions during the seven days of EMA, resulting in missing data for bridging, bonding, and stressful interaction measures. Consequently, the sample size for models including these variables was smaller than for models using other social interaction measures. This raised the possibility that differences in results across models could be due to sample size variation. To address this, we conducted a sensitivity analysis using a consistent sample across all models (N = 261). As shown in Figure 2, results remained substantively unchanged, suggesting that differences in sample size did not bias the main findings.

We also conducted a sensitivity analysis by modeling multiple measures of social interaction simultaneously. Because EMA-based quantitative measures—such as the proportion of time spent alone, time spent socializing, and the mean number of interaction partners—were highly correlated, we included only the proportion of time spent alone in the model alongside network size and qualitative measures of social interaction. As shown in Figure 3, the results were not substantially different.

## 4. Discussion

This study provides real-time, ecologically valid evidence that the quality of daily social interactions—rather than their quantity—is significantly associated with loneliness among older adults. Using EMA, we found that emotionally meaningful interactions (i.e., bonding social capital) and the absence of social strain were linked to lower levels of chronic loneliness. In contrast, quantitative indicators such as the proportion of time spent alone or socializing and the number of interaction partners were not significantly related to loneliness.

The null findings for the EMA-based quantitative measures in our study contrast with prior survey-based research that has found significant associations between social isolation, contact frequency, and loneliness [5,17,33], which is replicated in our significant association between network size and loneliness. One potential explanation for this discrepancy lies in how social isolation and contact frequency are typically measured in surveys. These studies often rely on participants’ reported number of close friends or frequency of contact with family members and other strong ties [5,17,33]. While focusing on close relationships helps reduce participant burden and recall bias—especially when asking about interactions over extended periods [37,38]—this approach inherently filters for relationship quality. As a result, it becomes difficult to disentangle the effects of interaction frequency from the emotional context of those interactions. By contrast, EMA collects real-time data on all social interactions, capturing interactions across the full spectrum of relationship types and contexts. This approach provides a more objective assessment of contact frequency, avoids conflating quantity with quality, and allows for a fine-grained examination of how daily social experiences relate to loneliness.

These findings challenge the conventional assumption that more frequent social contact necessarily protects against loneliness. It underscores the limitations of using frequency of contact and social isolation as standalone indicators of emotional well-being. While these metrics are often used in public health assessments and aging research, our results suggest that they may fail to capture the affective and relational depth that truly buffers against loneliness. Simply being around others does not necessarily reduce loneliness [39].

Among the qualitative dimensions, bonding social capital emerged as a strong protective factor. This aligns with prior research emphasizing the importance of emotionally supportive, trusting, and close relationships in reducing loneliness [28,33]. Our study extends this literature by showing that moment-to-moment social experiences of bonding also protect against chronic loneliness. This supports a growing body of evidence suggesting that the quality of interactions matters more than their mere occurrence [33,39].

Conversely, frequent stressful interactions were associated with higher loneliness. This finding highlights that not all social contact is beneficial—negative interactions can exacerbate feelings of isolation, possibly by reinforcing social withdrawal or eroding perceived social support [28,29]. This finding underscores the importance of assessing both positive and negative relational experiences.

Interestingly, bridging social capital was not significantly associated with loneliness. While bridging social capital may provide access to new perspectives or cognitive stimulation [24,26], our findings suggest that such interactions may not be emotionally fulfilling enough to reduce chronic loneliness. Alternatively, it may be that bridging encounters, which often occur with weak ties or in formal settings, lack the intimacy needed to combat loneliness.

To contextualize the effect size of social interactions, the observed standardized coefficient of 0.26 indicates a modest association between stressful interactions and loneliness. While this effect size is smaller than that observed for depressive symptoms (β = 0.59) in the same sample, a well-established correlate of loneliness, it nevertheless suggests that social interactions play a meaningful role in shaping loneliness levels.

Taken together, these findings suggest that efforts to reduce loneliness in later life should prioritize the cultivation of emotionally meaningful and low-conflict relationships rather than simply increasing social contact. Moreover, the use of EMA in this study demonstrates its value in capturing the nuances of daily social life that are often missed in retrospective surveys. Our findings also have important implications for the design of interventions, social services, and active aging policies. They suggest that efforts to reduce loneliness among older adults should move beyond simply increasing contact frequency and instead focus on improving the emotional quality of relationships. Interventions that foster emotionally supportive and low-strain interactions—such as peer support groups, intergenerational programs, or structured activities that promote trust and reciprocity—may be especially effective. At the policy level, community initiatives that expand opportunities for meaningful social engagement in both private and public settings can help strengthen the quality of older adults’ social networks. By emphasizing not just the quantity but the quality of social interactions, programs and policies can more effectively promote social well-being and reduce loneliness in later life.

### Limitations and Future Directions

Several limitations should be noted. First, the cross-sectional design of this study precludes causal inference. Although we hypothesize that the quality of social interactions influences chronic loneliness, it is also possible that individuals who have felt lonely for a long time perceive their social encounters more negatively [40,41]. Future longitudinal studies are needed to examine potential reverse causality. Second, while our sample was probability-based and state-representative, the generalizability of findings beyond Indiana may be limited. Third, our EMA prompts were administered only between 8:00 a.m. and 8:00 p.m., potentially missing meaningful social interactions occurring outside this timeframe. Fourth, the present study assessed chronic loneliness, which may not capture fluctuations in momentary feelings of loneliness. Future research should incorporate EMA of loneliness to better understand how daily social interactions relate to immediate emotional experiences in real time. Finally, future research should examine whether the associations observed here differ across demographic subgroups. For example, gender and racial differences in the meaning, context, and accessibility of social interactions could shape the relationship between social experiences and loneliness [42,43]. While our current sample (N = 272) is too small to conduct stratified or interaction analyses, investigating such heterogeneity in larger samples would provide valuable insights and help inform more culturally responsive interventions.

## 5. Conclusions

This study advances our understanding of loneliness in later life by demonstrating that the emotional quality of social interactions—rather than their frequency—is key to reducing chronic loneliness. EMA offers a powerful lens for capturing the social dynamics in a daily setting. Future research could complement these insights by examining additional factors such as social network structure, physical health, or cognitive functioning, which may interact with social experiences to influence loneliness more broadly. Interventions aimed at fostering emotionally supportive and low-strain relationships may be particularly promising for promoting loneliness and mental health among older adults.

## Figures and Tables

**Figure 1 ijerph-22-01411-f001:**
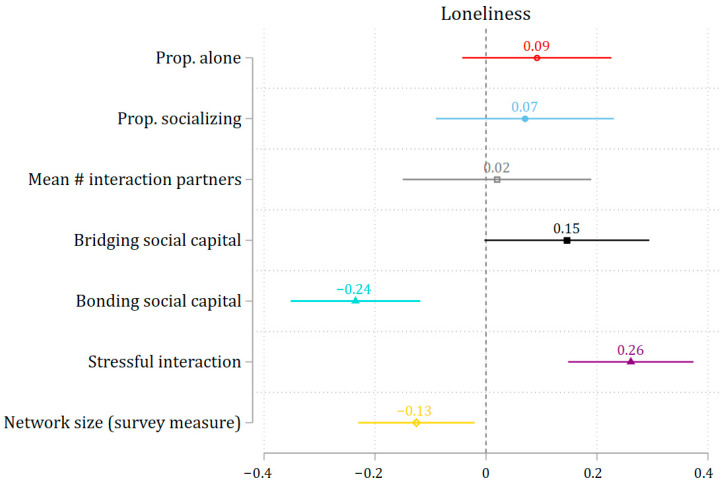
The association between social interactions and loneliness. Note: 95% confidence interval. All social interaction measures are standardized to enable comparison of effect sizes.

**Figure 2 ijerph-22-01411-f002:**
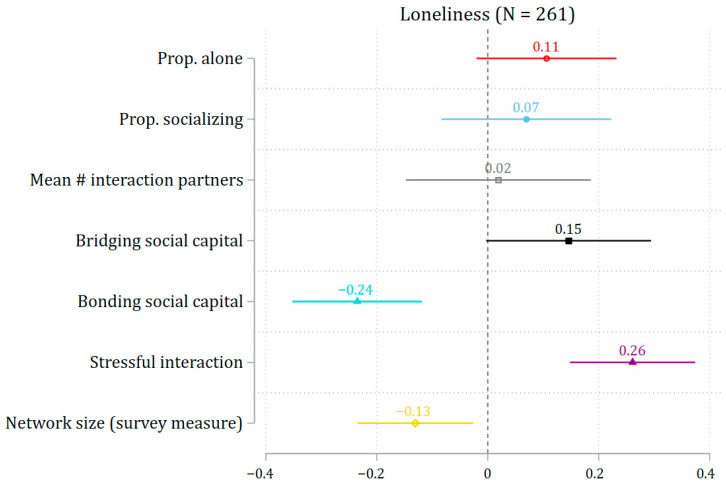
The association between social interactions and loneliness (the same sample size). Note: 95% confidence interval. All social interaction measures are standardized to enable comparison of effect sizes.

**Figure 3 ijerph-22-01411-f003:**
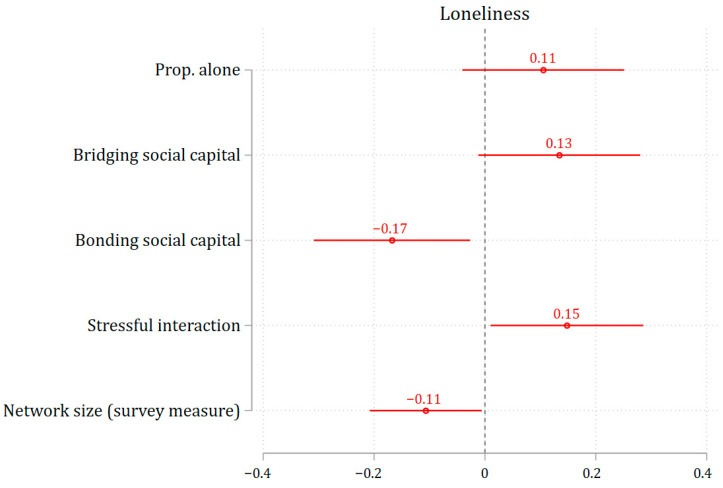
The association between social interactions and loneliness (simultaneously in the same model). Note: 95% confidence interval. All social interaction measures are standardized to enable comparison of effect sizes.

**Table 1 ijerph-22-01411-t001:** Descriptive statistics.

	Mean/Proportion	SD	Range	n
Women	0.51	0.46		272
Age	66.10	7.70	55–88	272
White	0.87	0.28		272
* Education *				272
Less than HS	0.05			
HS or GED	0.27			
Some college	0.24			
College	0.43			
Employed	0.40	0.48		272
Functional activity limitation	0.48	0.50		272
Network size	5.02	2.81	1–19	272
Loneliness	1.46	0.62	1.00–3.00	272
**EMA measures**				
Prop. alone	0.58	0.21	0–1	272
Prop. socializing	0.09	0.09	0–0.57	272
Mean # interaction partners	0.93	0.73	0–5	272
Bridging social capital	4.87	1.83	1.00–9.42	261
Bonding social capital	7.47	1.31	3.20–10.00	261
Stressful interaction	2.49	1.35	1.00–8.00	261

Note: Prop. = Proportion; HS = High school; GED = General educational development; SD = Standard deviation; EMA = ecological momentary assessment.

## Data Availability

SECHURA replication data can be requested from the corresponding author.

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
