# Peer review of "Quality over Quantity: The Association Between Daily Social Interactions and Loneliness"

_ijerph, 2025, doi:10.3390/ijerph22091411_

Round 1

Reviewer 1 Report

Comments and Suggestions for Authors

This article is well-structured, clear and addresses an extremely pertinent topic within public health and gerontology. The methodology, particularly the use of Ecological Momentary Assessment (EMA), is appropriate, and allows for multiple precise evaluations of the outcome in question. The results are compelling and support the central hypothesis. Although the manuscript is of high quality and well-structured, the Authors may find few minor observations below.

Minor observations:

  • Abstract: A minor enhancement could be to explicitly mention the probability-based sample from Indiana to underscore the representativeness of the data.
  • Methods: To ensure full transparency and confirm adherence to ethical research standards, it would be beneficial for the Methods section to explicitly state whether the study was conducted in accordance with the Declaration of Helsinki. Furthermore, details regarding the ethics committee approval, including the name of the approving committee and the protocol number, could be included, as well as information regarding preregistered protocol.
  • Methods, Sample, line 93: There is an inaccuracy in reporting the acronym.
  • Methods, Sample: According to the STROBE Statement (doi: 10.1016/j.jclinepi.2007.11.008), participant numbers (509 for survey, 272 for EMA), response rate (61%), and EMA compliance rate (76%) represent a result of the research and should be therefore moved to the appropriate section.
  • Methods, Analytic Strategy: This section should include the criterion for statistical significance. Moreover, the data distribution (e.g., normality) needs to be clarified.
  • Methods, Analytic Strategy: Analysing each social interaction variable in separate models is reasonable, but this approach may limit insights into their joint effects. It would strengthen the paper to briefly justify this choice or include supplementary models with multiple predictors.
  • Results, Table 1: For categorical variables (e.g., "Female", "White"), which are presently shown with means and standard deviations, it would enhance the interpretability of the sample distribution if absolute frequencies (n) and/or percentages (%) were provided.
  • Discussion: It would be beneficial to include a brief comment regarding the magnitude of the observed effects, particularly in relation to the clinical or practical significance of the identified associations, even if some effect sizes are small. Moreover, but an additional sentence emphasising the need for longitudinal studies to explicitly explore reverse causality would further strengthen this section.

Reviewer 2 Report

Comments and Suggestions for Authors

Dear Authors

The subject studied certainly has interest. Scientific evidence has shown that both the quantity and quality of interactions and relationships established between elderly people and those around them are associated with their well-being and influence their health/mental health. The proposal submitted, from a structural point of view, meets the established requirements and globally constitutes itself as a scientific article with an opportunity to provoke a shared analysis with the scientific community and with readers on the subject addressed. The text is, overall, logical and presents coherence among the different chapters. The scientific language is appropriate.

It also gathers the necessary ethical conditions for research with human beings approved by a referenced scientific committee.

The title is objective, highlighting both the quantity and quality of interactions between the elderly and those around them regarding their well-being and health.

Abstract - Concise, highlighting key ideas and main results.

The study specifically investigated how different types of everyday social interactions are associated with chronic loneliness in older people. The focus of your investigation was on how aggregate measures of daily social contact are associated with chronic or trait loneliness, as measured by the UCLA Loneliness Scale during an in-person survey, rather than with momentary loneliness assessed through ecological momentary assessment (EMA). We achieve this by combining data that were collected via an in-home survey and an EMA module to examine and compare how a broad range of moment-to-moment social experiences relate to perceptions of chronic loneliness in later life. The main conclusion highlighted is the association found between the quality of daily interactions and loneliness.

Keywords - Well selected.

Introduction - frames the study from a theoretical and scientific perspective, informing the objectives of the authors. It references a pertinent set of scientific references positioning the readers regarding the 'state of the art' on the subject under study. The main question, which is the focus of the study, is mentioned diluted in the text of this chapter.

Method - A transversal study with properly identified variables.

The sample selection was made from a duly referenced and credible source. The criteria on which the sample size and selection were based are not clarified. Could you please comment?

It will also be important to understand why the sample of your research consisted only of elderly people living at home. Thank you.

The variables are properly operationalized.

The research instruments are outlined and the main psychometric characteristics are referred to, which is a very positive aspect. The method chosen for data collection has specific characteristics.

Could you please clarify how the effective participation of the selected elderly persons was ensured? Were in-person contacts made?

The analytical treatment of the collected data is explicit and demonstrates your attention to making the robustness of its application understandable.

Some details about the measures and tests used for this purpose are presented.

Results - Is it possible to clarify what results were achieved through the application of the UCLA to elderly people? Or is this determination not relevant in relation to the study's objectives? I may not have fully understood. However, the reader might be interested in this issue. The presentation of the main results is well supported, with the appropriate description and illustration through the tables and figures, which are well-structured and illustrative. Their placement in the text is very balanced.

Discussion – Logical sequence with interpretations and confrontations with the previously presented pertinent evidence. Consistent with the results obtained and presented. I suggest a bit more depth, particularly regarding the methodology and its applicability with advantages, and the main results obtained in confrontation with the prior evidence. The limitations subsection denotes very positive critical reflection and analysis.

Conclusions – it is important to specify this chapter. However, perhaps they could reconsider some enhancement of it, highlighting its usefulness and suggesting other variables that could be studied complementarily for a deeper understanding of the subject.

 References – well configured.

Best regards.

Reviewer 3 Report

Comments and Suggestions for Authors

The article “Quality over Quantity: The Association between Daily Social Interactions and Loneliness” investigates the relationship between daily social interactions and loneliness in older adults, using Ecological Momentary Assessment (EMA) data. The topic is of considerable scientific relevance for public health, and the methodological approach is innovative and timely. The study's main contribution lies in reinforcing the importance of the quality of social interactions (specifically bonding social capital and the absence of stressful encounters) over the mere quantity of contact, a conclusion with important implications for active aging policies.

The article is well-structured, following the standard format of an empirical article, with clear sections for the introduction, methods, results, and discussion. The overall coherence between the research problem, the literature review, and the methodology is strong.

Although the text shows considerable promise, there are several crucial areas that require revision before it can be considered for publication. The comments I mention below are intended to help the authors rethink some issues:

1. The introduction effectively contextualizes the study and defines its objectives, but it fails to formulate clear, a priori research hypotheses. A specific section should be included at the end of the literature review presenting the hypotheses to be tested. For example: H1: A higher level of bonding social capital is negatively associated with chronic loneliness.

2. It seems to me that there is a significant conceptual problem in the mismatch between the predictor variables and the outcome variable. The study relates momentary social interactions (measured via EMA) to a measure of chronic loneliness (the UCLA Loneliness Scale). This temporal difference creates a misalignment, a limitation that the authors themselves acknowledge. The study design would be considerably more robust if it included a momentary measure of loneliness based on EMA, which would allow for a time-aligned analysis of how specific interactions influence feelings of loneliness in real-time.

3. The construct validity for "bridging social capital" is questionable. The items used (e.g., “being exposed to new information”) seem to capture cognitive stimulation more than the social connections that define this form of capital. This potential mismatch in operationalization could explain the absence of a significant association with loneliness. The authors should provide a stronger theoretical justification for their choice of items or reconsider this operationalization.

4. The article would be substantially strengthened by exploring potential sociodemographic differences. The authors mention that factors such as gender or race/ethnicity could influence the results, but they do not test these possibilities. Given the importance of these variables in the literature on loneliness and social networks, conducting stratified or interaction analyses (e.g., by gender, race, socioeconomic status) is a fundamental step to increase the study's contribution.

Minor Points for Revision

5. Although Figures 1-3 are informative, they present overlapping results. To improve conciseness and avoid redundancy, the authors could consider condensing them into a single figure with multiple panels.

6. The literature review is solid, balancing classic sources with recent publications. However, it is heavily focused on the US context. Dialogue with international comparative literature, especially from European contexts where loneliness and aging are also central public policy issues, would broaden the study's scope and relevance.

7. The results have clear applied relevance. The discussion would benefit from a more detailed exploration of how these results can inform the design of specific interventions, social services, and active aging policies aimed at improving the quality of social networks for the elderly.

This is a valuable and well-conducted study with a strong methodological foundation. However, the aspects we have identified above should be improved for the sake of the research itself.

Thus, my recommendation is to Accept with Major Revisions. I am confident that if the authors can satisfactorily address these points, the text will represent a significant contribution to the field.

Round 2

Reviewer 3 Report

Comments and Suggestions for Authors

Thank you for your thorough revisions to the manuscript “Quality over Quantity: The Association between Daily Social Interactions and Loneliness.” The addition of clear hypotheses (H1-H3), robust theoretical justification for bridging social capital, inclusion of international literature, and a detailed discussion on policy implications have significantly strengthened the paper. The acknowledgment of limitations, supported by the comprehensive Table S1 (e.g., β=-0.235*** for bonding social capital, β=0.261*** for stressful interactions), adds transparency and credibility.

Given the substantial improvements, the manuscript is now suitable for publication in its present form.